# OpenReview forum: "DRIFT-BENCH: Diagnosing CoopeRative Breakdowns in LLM Agents under Input Faults via Multi-Turn Interaction"
_ICML.cc/2026/Conference — ICML 2026 regular_

### Official Review · Reviewer_6Rrd · 2026-02-17

**Soundness:** 2
**Presentation:** 2
**Significance:** 2
**Originality:** 2
**Overall Recommendation:** 3
**Confidence:** 3

**Summary:**

DRIFT-BENCH is the first diagnostic benchmark designed to evaluate LLM agent resilience against systematic cooperative breakdowns. The benchmark assesses agent performance across two categories of environments: State-Oriented Environments (Operating Systems and Databases) and Service-Oriented Environments (APIs). To simulate realistic and diverse human interaction during evaluation, DRIFT-BENCH incorporates a User Simulator built around four distinct persona types.

**Compliance With Llm Reviewing Policy:**

Affirmed.

**Final Justification:**

In light of the efforts given by the authors for the rebuttal I have raised scores (soundess, significance, presentation) to fair from poor. The overall has been changed to weak reject from reject.

**Key Questions For Authors:**

The appendix provides valuable and detailed information that supports the main text. However, it would benefit from further distillation and organization, as its current length makes it somewhat difficult to navigate and potentially distracting for readers.

**Limitations:**

Yes

**Strengths And Weaknesses:**

Invoking Grice and Austin feels superficial. The taxonomy could have been defined without citing communication theory.
The environments are not novel (OS tasks, APIs). The LLM-as-user simulator is with personas are heuristic. It is obvious that the models degrade but there is little analysis as to what are the causes and how to overcome them.

---

> ### Author Rebuttal · Authors · 2026-03-28
>
> We sincerely thank you for your time and hope the following clarifications and new experiments address the concerns.
>
> ---
>
> ## W1: Theoretical Grounding
>
> > *"Invoking Grice and Austin feels superficial."*
>
> A key motivation is **fragmentation in existing literature**: NoisyToolBench covers premise/expression faults under idiosyncratic names; IN3 addresses intention faults in isolation; CLARQ, CLAMBER, AmbigQA, CondAmbigQA each handle expression-level ambiguity with incommensurable definitions (Table 1). Cross-benchmark comparison is impossible.
>
> Our taxonomy serves as a **unification of these scattered efforts**. Mapping onto Grice's maxims (Relation → Intention, Quality → Premise, Quantity → Parameter, Manner → Expression) provides: (1) a **completeness argument** — four theoretically exhaustive dimensions, not arbitrary, and (2) a **common coordinate system** for existing and future benchmarks. This is consolidation, not decorative citation.
>
> ---
>
> ## W2: Novelty of Environments
>
> > *"The environments are not novel (OS tasks, APIs)."*
>
> We hope to clarify a possible misunderstanding. OS/API tasks are application scenarios; DRIFT-BENCH's contribution is the **full pipeline**: upstream data construction (semantic frame extraction → perturbation → fault injection), the interactive layer (persona-driven simulator × clarification strategies), and downstream evaluation (RISE protocol). None exist in the original AgentBench or StableToolBench. Established environments ensure performance changes are attributable to our faults, not task difficulty.
>
> ---
>
> ## W3: User Simulator
>
> > *"The LLM-as-user simulator with personas is heuristic."*
>
> **Human evaluation (Appendix D):** Two annotators labeled 198 responses; Cohen's κ = 0.7649, annotator-vs-GT accuracy = 86.87%. **Leakage control:** Prompt enforces "NEVER make up new information"; simulator never receives ground-truth answers. **Downstream signal:** Persona-dependent variation in Table 5 (Avoidant: 56.64% vs. Spontaneous: >67%); Spontaneous–Avoidant r = 0.138, confirming orthogonal styles. All multi-turn agent benchmarks (τ-Bench, τ²-Bench) use LLM-simulated users; our GDMS-grounded personas with empirical validation go beyond prior approaches.
>
> ---
>
> ## W4: Causal Analysis and Mitigation
>
> > *"There is little analysis as to what are the causes and how to overcome them."*
>
> We have conducted extensive new analyses beyond our error study (Appendix F):
>
> ### A. Failure Mode Attribution (1,018 pairs, service-oriented)
>
> | Model             | Degrad. Rate | Dominant Mode          | Attribution |
> | ----------------- | ------------ | ---------------------- | ----------- |
> | GPT-5.2 / GLM-4.7 | <10%         | —                      | Minimal     |
> | GPT-OSS-120B      | 30.0%        | **Syntactic Collapse** | **66.7%**   |
> | Qwen3             | 52.7%        | **Abandonment**        | **62.0%**   |
> | Gemini-2.5-Flash  | 73.3%        | **Abandonment**        | **90.0%**   |
> | Llama-4           | 82.0%        | **Clarif. Loop**       | **73.2%**   |
>
> Three distinct mechanisms, not a single "context overload."
>
> ### B. Information Injection Ablation
>
> | Model         | Fault      | NoClarify | Clarify | G      | Repaired | G      |
> | ------------- | ---------- | --------- | ------- | ------ | -------- | ------ |
> | GPT-OSS-120B  | Intention  | 26.83     | 32.52   | +5.69  | 36.59    | +9.76  |
> |               | Premise    | 37.40     | 49.59   | +12.19 | 41.46    | +4.06  |
> |               | Parameter  | 24.39     | 58.54   | +34.15 | 55.29    | +30.90 |
> |               | Expression | 33.34     | 43.09   | +9.75  | 51.22    | +17.88 |
> | DeepSeek-V3.2 | Intention  | 32.52     | 40.65   | +8.13  | 48.78    | +16.26 |
> |               | Premise    | 41.46     | 53.66   | +12.20 | 51.22    | +9.76  |
> |               | Parameter  | 30.89     | 59.35   | +28.46 | 53.66    | +22.77 |
> |               | Expression | 40.65     | 50.41   | +9.76  | 56.10    | +15.45 |
>
> Communicative faults: Repaired > Clarify. Structural faults: Clarify > Repaired.
>
> ### C. Compound Faults
>
> | Model         | Intent. | +Param. | PD   | Prem. | +Intent. | PD   | Param. | +Expr. | PD   | Expr. | +Prem. | PD   |
> | ------------- | ------- | ------- | ---- | ----- | -------- | ---- | ------ | ------ | ---- | ----- | ------ | ---- |
> | GPT-OSS-120B  | 32.52   | 23.33   | -28% | 49.59 | 15.86    | -68% | 58.54  | 21.95  | -63% | 43.09 | 19.52  | -55% |
> | DeepSeek-V3.2 | 40.65   | 28.05   | -31% | 53.66 | 20.73    | -61% | 59.35  | 31.71  | -47% | 50.41 | 23.17  | -54% |
>
> 28–68% additional degradation; single-fault results are conservative lower bounds.
>
> ### Mitigation
>
> Our **clarification skill set** (§3.2) yields up to +22.59pp recovery (Table 3). Findings further suggest risk-aware prompting, context pruning, and environment-aware policies.

---

> > ### Author Rebuttal · Reviewer_6Rrd · 2026-04-04
> >
> > My questions and concerns are fully addressed.

---

### Official Review · Reviewer_eYUd · 2026-02-18

**Soundness:** 3
**Presentation:** 3
**Significance:** 4
**Originality:** 3
**Overall Recommendation:** 5
**Confidence:** 3

**Summary:**

The authors introduce DRIFT-BENCH, a benchmark for evaluating models under "imperfect" input. Furthermore, they propose the rise valuation framework to test the agent's robustness, intelligence, safety, and efficiency. For state-oriented tasks, they observe a significant performance degradation across the board, with some more moderate degradation.
They conduct a systematic evaluation of different models when interacting with different "user types" and detect a significant drop in performance for frontier models and demonstrate that the models often even proceed with high-risk actions.
The benchmark ensures that the tasks are solvable, taking into account the need for clarification and the five different personas.

**Compliance With Llm Reviewing Policy:**

Affirmed.

**Final Justification:**

Following the rebuttal process, I remain committed to my recommendation of accepting the paper.

**Key Questions For Authors:**

None

**Limitations:**

yes

**Strengths And Weaknesses:**

Soundness: Paper is sound, the claims and analysis are backed up by empirical results.
Presentation: Paper is well written and organized.
Significance: The paper is significant, as it addresses the very important issue of imperfect input.
Originality: The work is original as they introduce a new needed benchmark.

---

> ### Author Rebuttal · Authors · 2026-03-28
>
> We sincerely thank you for your time, effort, and the highly positive evaluation of our work.
>
> We are greatly encouraged that you recognize the critical importance of addressing "imperfect" user inputs in agent evaluations. We are also thrilled that you find our proposed DRIFT-BENCH and the RISE evaluation framework to be sound, original, and well-presented. Your acknowledgement of our systematic evaluation design—particularly the inclusion of different user personas and the demonstration of models proceeding with high-risk actions without clarification—validates the core motivation of our benchmark.
>
> Thank you again for your strong support and for championing the significance of our research!

---

> > ### Author Rebuttal · Reviewer_eYUd · 2026-04-01
> >
> > I already recommended accepting the paper

---

### Official Review · Reviewer_P3S6 · 2026-03-10

**Soundness:** 4
**Presentation:** 4
**Significance:** 3
**Originality:** 3
**Overall Recommendation:** 6
**Confidence:** 4

**Summary:**

This paper addresses a critical gap in evaluating LLM-based autonomous agents. As LLMs increasingly act as agents that execute real-world tasks (manipulating files, calling APIs, running code), user inputs often contain "cooperative faults" — implicit intentions, false presuppositions, missing parameters, or ambiguous expressions — that create execution risks existing benchmarks fail to capture. The authors introduce DRIFT-BENCH, the first benchmark designed to evaluate how agents handle such input faults through multi-turn clarification across both state-oriented and service-oriented environments. Drawing on classical communication theory (Grice's Cooperative Principle and Austin's speech-act theory), the benchmark defines a unified taxonomy of cooperative breakdowns across four categories — Flaw of Intention, Flaw of Premise, Flaw of Parameter, and Flaw of Expression — and uses a persona-driven user simulator alongside the RISE evaluation protocol, which measures agent performance across four axes: Robustness, Intelligence, Safety, and Efficiency. Experiments on a range of frontier models (GPT-5.2, Gemini, Qwen3, Deepseek, Llama, etc.) reveal substantial performance drops under faulty inputs, with agents proceeding with high-risk actions in ~70% of cases rather than seeking clarification, highlighting a systemic failure in agentic safety that DRIFT-BENCH is designed to systematically diagnose.

**Compliance With Llm Reviewing Policy:**

Affirmed.

**Final Justification:**

My questions are fully resolved! Thank you very much!

**Key Questions For Authors:**

see weakness above

**Limitations:**

yes

**Strengths And Weaknesses:**

1. This paper investigates an important problem where inputs may be faulty and requires clarifications to fix
2. The study is comprehensive. The authors categorize the flaw into various perspectives in intention, parameter, expression, and analyze results from reliability, interaction, safety and efficiency; the evaluation covers both proprietary and open-source models

Weakness:
1. The perturbation seems to be artificially created by authors, which may differ from the realistic scenarios.
2. I am wondering what approaches are used to ensure the quality of user simulator, i.e., natural, realistic, not provide too much information, etc.

---

> ### Author Rebuttal · Authors · 2026-03-28
>
> We thank you for the positive assessment and thoughtful questions. We really appreciate your recognition. We have carefully addressed each of your concerns below.
>
> ---
>
> ## Weakness 1: Ecological Validity of Perturbations
>
> > *"The perturbation seems to be artificially created by authors, which may differ from the realistic scenarios."*
>
> Thank you for raising this concern. DRIFT-BENCH adopts controlled single-fault perturbations for diagnostic attribution — isolating fault types is necessary to determine *which* category causes a given failure. To test whether findings generalize to more realistic scenarios where multiple faults co-occur, we conducted **compound fault experiments** that overlay several randomly combined faults on state-oriented (OS) tasks under the w/ Clarify condition.
>
> ### Results
>
> **Table R1.** Single-fault vs. compound-fault accuracy (w/ Clarify, state-oriented OS). PD = relative drop from single to compound.
>
> | Model         | Intent. | +Param. | PD     | Prem. | +Intent. | PD     | Param. | +Expr. | PD     | Expr. | +Prem. | PD     |
> | ------------- | ------- | ------- | ------ | ----- | -------- | ------ | ------ | ------ | ------ | ----- | ------ | ------ |
> | GPT-OSS-120B  | 32.52   | 23.33   | -28.3% | 49.59 | 15.86    | -68.0% | 58.54  | 21.95  | -62.5% | 43.09 | 19.52  | -54.7% |
> | DeepSeek-V3.2 | 40.65   | 28.05   | -31.0% | 53.66 | 20.73    | -61.4% | 59.35  | 31.71  | -46.6% | 50.41 | 23.17  | -54.0% |
>
> Average compound PD: GPT-OSS-120B = **-53.4%**, DeepSeek-V3.2 = **-48.2%**.
>
> ### Analysis
>
> 1. **Compound faults cause catastrophic additional degradation** (28–68% further drop), confirming that single-fault results are a conservative lower bound on real-world difficulty.
>
> 2. **The severity ordering persists**: premise-involving combinations remain most damaging under compound conditions, validating that single-fault diagnostics reliably identify the most vulnerable categories.
>
> 3. **Diagnostic isolation remains essential**: under the Premise+Intention compound (GPT-OSS-120B: 15.86%), it is impossible to attribute the failure without separate single-fault baselines. This justifies DRIFT-BENCH's design.
>
> Additionally, our human evaluation (Appendix D, Cohen's κ = 0.76, Accuracy_A vs GT = 86.87%) confirms that both perturbations and persona-driven responses are natural and distinguishable by human annotators.
>
> ---
>
> ## Weakness 2: User Simulator Quality Assurance
>
> > *"I am wondering what approaches are used to ensure the quality of user simulator."*
>
> We employ multiple mechanisms:
>
> **Naturalness and persona fidelity.** Our human evaluation (Appendix D) had two annotators independently label 198 simulator responses by persona type. Inter-annotator agreement was high (Cohen's κ = 0.7649), and annotator-vs-ground-truth accuracy reached 86.87%, confirming that simulator outputs are distinguishable and consistent with designed personas.
>
> **Information leakage control.** The simulator prompt (Appendix H.3) enforces strict rules: (1) "NEVER make up new information or change your original intent"; (2) "Do not provide new details beyond what's in your original intent." The simulator receives the original intent and perturbed description — never the ground-truth answer — preventing direct leakage.
>
> **Behavioral validity via downstream signal.** Table 5 shows significant persona-dependent performance variation (e.g., Avoidant: 56.64% avg. vs. Spontaneous: >67%). If the simulator lacked fidelity or distinctiveness, such systematic differences would not emerge. The near-zero Pearson correlation between Spontaneous and Avoidant personas (r = 0.138, Table 13) further confirms that these represent genuinely orthogonal interaction styles.
>
> **Multi-model rotation.** Each case is assigned a consistent LLM for simulation (§3.3), preventing single-model artifacts from dominating the results.

---

> > ### Author Rebuttal · Reviewer_P3S6 · 2026-04-01
> >
> > My questions are fully resolved, and I raised my rating!

---

> > > ### Author Response · Authors · 2026-04-01
> > >
> > > Thank you very much for the raised score and your constructive feedback!
> > > We are pleased that our response have addressed your concerns and strengthened the overall impact of the paper.

---

### Official Review · Reviewer_DZ5S · 2026-03-12

**Soundness:** 3
**Presentation:** 3
**Significance:** 3
**Originality:** 3
**Overall Recommendation:** 3
**Confidence:** 4

**Summary:**

This paper studies how large language model (LLM) agents handle faulty or incomplete user inputs in multi-turn interactions. The authors introduce DRIFT-BENCH, a benchmark designed to evaluate an agent’s ability to diagnose and recover from cooperative breakdowns during interaction. The paper further proposes the RISE evaluation protocol, which measures agent performance along four dimensions: Robustness, Intelligence, Safety, and Efficiency.

**Compliance With Llm Reviewing Policy:**

Affirmed.

**Final Justification:**

I will keep my score after checking the whole rebuttal process

**Key Questions For Authors:**

1. Could the authors provide additional analysis or controlled experiments to isolate the effects of context overload? For example, have you tested whether controlling context length or tool-call formatting changes the observed behavior?
2. The paper focuses on diagnosing agent failures through the proposed benchmark but does not explore methods for mitigating these failures. Could the authors discuss potential approaches or provide preliminary experiments on how agents might be improved to better handle input faults?

**Limitations:**

See weekness

**Strengths And Weaknesses:**

Strength:

1. The proposed DRIFT-BENCH introduces a clear taxonomy of cooperative breakdowns, providing a systematic way to study different types of user–agent miscommunication.

Weekness:

1. The benchmark constructs input faults by applying controlled perturbations to existing tasks. Such perturbations may not fully reflect the complexity of real-world user interactions, where ambiguities, evolving goals, and contextual misunderstandings often arise simultaneously.

2. The paper attributes the observed “Clarification Paradox” to several factors, including context overload, syntactic collapse in tool calls, and abandonment. However, these explanations are mainly based on qualitative observations of interaction logs rather than controlled experiments that isolate and verify each factor. For example, the paper does not systematically control context length, tool interface constraints, or agent policies to evaluate whether these factors independently cause the observed performance degradation.

---

> ### Author Rebuttal · Authors · 2026-03-28
>
> We thank your for your time and constructive feedback. We address each concern with new experiments and quantitative analyses.
>
> ---
>
> ## Weakness 1
>
> > *"Such perturbations may not fully reflect the complexity of real-world user interactions..."*
>
> We conducted compound fault experiments (detailed in our response to **Reviewer P3S6, Table R1**). Pairwise fault combinations cause an additional 28–68% relative drop on top of single-fault degradation, with average compound PD of -53.4% (GPT-OSS-120B) and -48.2% (DeepSeek-V3.2). The severity ordering from single-fault diagnostics persists, confirming that single-fault results are a conservative lower bound and that diagnostic isolation is necessary for causal attribution.
>
> ## Weakness 2
>
> > *"These explanations are mainly based on qualitative observations... does not systematically control..."*
>
> ### A. Service-Oriented: Failure Mode Attribution
>
> We analyzed **1,018 matched query pairs** on service-oriented tasks, classifying each degraded case:
>
> - **Group A — Syntactic Collapse**: More `Tool input parse error` events under clarify (addresses **tool-call formatting**).
> - **Group B — Abandonment Catalyst**: Agent issues `give_up_and_restart` under clarify but not no-clarify.
> - **Group C — Clarification Loop** (new): Tool call count decreases without parse errors or give-ups — dialogue *replaces* execution.
>
> Overall: 404/1018 (39.7%) hurt by clarification, 64 (6.3%) helped. Per-model profiles:
>
> | Model             | Degrad. Rate | Dominant Mode | Key Attribution                                        |
> | ----------------- | ------------ | ------------- | ------------------------------------------------------ |
> | GPT-5.2 / GLM-4.7 | <10%         | —             | Minimal degradation                                    |
> | GPT-OSS-120B      | 30.0%        | **Group A**   | **66.7% Syntactic Collapse** (+4.8 parse errors/query) |
> | Qwen3             | 52.7%        | **Group B**   | **62.0% Abandonment**                                  |
> | Gemini-2.5-Flash  | 73.3%        | **Group B**   | **90.0% Abandonment**                                  |
> | Llama-4           | 82.0%        | **Group C**   | **73.2% Clarification Loop**                           |
>
> The Paradox stems from distinct, model-specific mechanisms — not a single "context overload" factor.
>
> ### B. Information Injection Ablation
>
> To isolate **context length effects**, we compare: **NoClarify**, **Full Clarify** (full history), and **Repaired Input** (clarification info injected as one sentence, no history). Sampled from both environments:
>
> | Model         | Fault      | NoClarify | Full Clarify | G      | Repaired | G      |
> | ------------- | ---------- | --------- | ------------ | ------ | -------- | ------ |
> | GPT-OSS-120B  | Intention  | 26.83     | 32.52        | +5.69  | 36.59    | +9.76  |
> |               | Premise    | 37.40     | 49.59        | +12.19 | 41.46    | +4.06  |
> |               | Parameter  | 24.39     | 58.54        | +34.15 | 55.29    | +30.90 |
> |               | Expression | 33.34     | 43.09        | +9.75  | 51.22    | +17.88 |
> | DeepSeek-V3.2 | Intention  | 32.52     | 40.65        | +8.13  | 48.78    | +16.26 |
> |               | Premise    | 41.46     | 53.66        | +12.20 | 51.22    | +9.76  |
> |               | Parameter  | 30.89     | 59.35        | +28.46 | 53.66    | +22.77 |
> |               | Expression | 40.65     | 50.41        | +9.76  | 56.10    | +15.45 |
>
> **Communicative faults** (intention, expression): Repaired > Clarify — dialogue is overhead once ambiguity is resolved. **Structural faults** (premise, parameter): Clarify > Repaired — interaction provides verification value beyond static injection.
>
> ### C. Synthesis
>
> Service-oriented environments degrade across *all* fault types (A). Dialogue helps only structural faults requiring verification (B). Opaque environments afford no state inspection, so dialogue becomes overhead — manifesting as abandonment, clarification loops, or syntactic collapse.
>
> ---
>
> ## Q1: Context Overload and Tool-Call Formatting
>
> Please see w2 for details.
>
> ## Q2: Mitigation Approaches
>
> DRIFT-BENCH is primarily diagnostic. We note that prior benchmarks focused on narrower input faults already have associated mitigation literature: AmbigQA, CondAmbigQA, and CLAMBER address expression-level ambiguity (a subset of our Expression category), and IN3, UserBench target implicit intentions. Mitigation techniques *exist* for individual fault types when evaluation settings are well-defined. What has been lacking is not methods but a unified diagnostic framework spanning the full fault taxonomy under grounded execution — precisely DRIFT-BENCH's contribution.
>
> Our benchmark already provides an initial mitigation: the **clarification skill set** (§3.2), yielding up to +22.59pp recovery (Table 3). Findings further suggest: risk-aware prompting, context pruning, and environment-aware policies routing fault types to appropriate resolution channels.

---

> > ### Author Rebuttal · Reviewer_DZ5S · 2026-04-01
> >
> > The rebuttal provides additional experiments that partially address the concerns, particularly through compound fault settings and the information injection ablation.
> >
> > For Weakness 1, the introduction of compound fault experiments is helpful and demonstrates that performance degradation compounds under multiple perturbations. However, this does not fully resolve the concern about realism, as a distributional analysis or comparison with real-world data would be important to validate whether these perturbations reflect realistic user interactions.
> >
> > For Weakness 2, while the information injection ablation provides useful insight into the role of context length versus information content, the overall explanation of the “Clarification Paradox” remains largely based on observational analysis. The failure mode attribution is descriptive rather than causal, and key factors such as tool interface constraints and agent policies are not systematically controlled or isolated. As a result, the causal validity of the proposed explanations remains only partially established.

---

> > > ### Author Response · Authors · 2026-04-02
> > >
> > > We sincerely thank you for the follow-up and the recognition that our additional experiments partially address the original concerns. We offer the following clarifications on the two remaining points.
> > >
> > > ---
> > >
> > > ### **Weakness 1 (distributional realism):**
> > >
> > > We fully agree that validating perturbations against real-world user-agent interaction data would be valuable. However, we want to highlight that the absence of such data is precisely the research gap that motivates DRIFT-BENCH. Large-scale, fault-annotated corpora of real user interactions with tool-using agents remain scarce — existing efforts in this space (e.g., NoisyToolBench, CLAMBER, IN3) similarly construct evaluation data by applying controlled perturbations to well-formed inputs, reflecting a shared methodological reality in the field. If it did, perturbation-based construction would be unnecessary, one could directly build a benchmark from naturalistic data. It is because this foundational resource is missing that we (1) ground our fault taxonomy in communication theory, which has long studied the space of human communicative errors, providing a principled proxy for what real faults look like; (2) adopt controlled perturbation as the construction method, following established practice in robustness research (e.g., adversarial NLP benchmarks universally rely on synthetic perturbations for the same reason); and (3) validate ecological plausibility indirectly through experimental signals — the substantial performance degradation under perturbation, the recovery when clarification skills are provided, and the compounding effect under multi-fault conditions all mirror expected real-world dynamics.
> > >
> > > We view DRIFT-BENCH as foundational infrastructure that makes distributional validation *possible* as a next step: the taxonomy defines what to annotate, the evaluation protocol defines how to measure, and the benchmark defines baseline expectations. Collecting and aligning real user data is important future work, but it builds on the framework we establish here, rather than replacing it.
> > >
> > > ---
> > >
> > > ### **Weakness 2 (causal validity):**
> > >
> > > We appreciate the reviewer's precision on this point. We want to clarify what our analysis claims and what it does not.
> > >
> > > Our failure mode attribution identifies systematic co-occurrences (e.g., clarification-enabled trajectories exhibit significantly more parse errors, or more frequent give-up events) and quantifies their prevalence across models. We have not claimed, nor attempted, strict interventionist causality in the sense of isolating a single mechanism while holding all others constant. We acknowledge this openly.
> > >
> > > However, we note that this standard is extremely difficult to meet in the context of LLM agent systems. Agent behavior emerges from opaque model internals — one cannot surgically disable "abandonment tendency" while keeping all other behaviors fixed, in the way one might knock out a gene in a biological experiment. Even in adjacent fields such as reinforcement learning, attributing policy failures (e.g., reward hacking, mode collapse) is done through descriptive and correlational analysis rather than strict causal intervention. The level of evidence we provide (quantified failure modes, per-model profiles, controlled ablation separating information content from dialogue process) is consistent with the analytical standards in current agent evaluation research.
> > >
> > > More importantly, establishing precise causal mechanisms for the Clarification Paradox is not our main contribution. Our quantitative analyses provide actionable hypotheses and diagnostic tools for the community to investigate these mechanisms further. We see the establishment of causal mechanisms and the development of targeted solutions as a natural next stage of research, one that our benchmark is specifically designed to support and accelerate. We are at the stage where the field needs the infrastructure to systematically observe and measure these phenomena, which is the gap DRIFT-BENCH fills. Causal investigation is a natural and important next step, enabled by the framework we provide.

---

### Decision · Program_Chairs · 2026-04-30

**Decision:**

Accept (regular)

**Comment:**

The paper proposes a benchmark called DRIFT-BENCH that is designed to study failures of multi-turn LLM agent systems in cooperative settings when exposed to faulty inputs. It measures propagation of errors propagate and degradation of collaboration between agents over time, and provides a structured testbed for diagnosing robustness issues in multi-agent interactions. Their results show that current SOTA LLM agents are still highly vulnerable to small perturbations in the inputs and they can cause cascading cooperative breakdowns.
Reviews suggest that the benchmark is well-motivated and impactful for agent evaluation. However, they also raise several concerns about the synthetic perturbations, the user simulator, and limited explanation of failure modes. After the rebuttals that include additional experiments, most of these concerns are resolved.